



# 1    Vertical profiles of sub-3 nm particles over the boreal forest

Katri Leino[1*], Janne Lampilahti[1], Pyry Poutanen[1], Riikka Väänänen[1], Antti Manninen[1], Stephany
Buenrostro Mazon[1], Lubna Dada[1], Anna Nikandrova[1], Daniela Wimmer[1], Pasi P. Aalto[1], Lauri R.
Ahonen[1], Joonas Enroth[1], Juha Kangasluoma[1], Petri Keronen[1], Frans Korhonen[1], Heikki Laakso[1],
Teemu Matilainen[1], Erkki Siivola[1], Hanna E. Manninen[1,2], Katrianne Lehtipalo[1], Veli-Matti
Kerminen[1], Tuukka Petäjä[1] and Markku Kulmala[1]
[1] Institute for Atmospheric and Earth System Research / Physics, Faculty of Science, P.O. Box 64, FI-00014 University
of Helsinki, Finland
(* corresponding author's email: katri.e.leino@helsinki.fi)
[2]CERN, CH-1211Geneva 23, Switzerland.
**Abstract.** This work presents airborne observations of sub-3 nm particles in the lower troposphere
and investigates new particle formation (NPF) within an evolving boundary layer (BL). We studied
particle concentrations together with supporting gas and meteorological data inside the planetary BL
over a boreal forest site in Hyytiälä, Southern Finland. The analysed data were collected during three
flight measurement campaigns: May-June 2015, August 2015 and April-May 2017, including 27
morning and 26 afternoon vertical profiles. As a platform for the instrumentation, we used a Cessna
aircraft. The analysed flight data were collected horizontally within a 30-km distance from the
SMEAR II station in Hyytiälä and vertically from 100 m above ground level up to 2700 m. The
number concentration of 1.5–3 nm particles was observed to be, on average, the highest near the
forest canopy top and to decrease with an increasing altitude during the mornings of NPF event days.
This indicates that the precursor vapours emitted by the forest play a key role in NPF in Hyytiälä.
During daytime, newly-formed particles were observed to grow in size and the particle population
became more homogenous within the well-mixed BL in the afternoon. During undefined days in
respect to NPF, we also detected an increase in concentration of 1.5–3 nm particles in the morning
but not their growth in size, which indicates an interrupted NPF process during these undefined days.
Vertical mixing was typically stronger during the NPF event days than during the undefined or non-
event days.




## 1 Introduction



One of the most important sources of secondary aerosol particles in the atmosphere is new particle
formation (NPF). NPF and subsequent growth is a globally observed phenomenon (Kulmala et al.,
2004; Kulmala and Kerminen, 2008; Kerminen et al., 2018). It is still partly unclear where, when and
how NPF occurs in the atmosphere. Aerosol measurements on board of an aircraft can give
information about the vertical, horizontal and spatial extent of the NPF in the lower atmosphere.
The planetary boundary layer (PBL) is a complex layer in the lowest part of the atmosphere, defined
as the part of the troposphere that is directly connected to the Earth's surface through the exchange
of momentum, heat and mass, and responds to surface forcing with a timescale of an hour or less
(Stull, 2012). The PBL has a characteristic diurnal cycle, but the detailed development varies from
day to day. Several meteorological, physical and chemical processes influence the spatial and
temporal conditions inside the BL, such as the boundary layer height (BLH) and mixing strength.
This gives rise to the complexity to define the exact BLH or to characterize the typical BL structure
or height at a given location.
Several airborne measurements have been conducted to investigate particle number concentrations
and size distributions as well as NPF inside the PBL. Over Europe, Crumeyrolle et al. (2010) observed
that the horizontal extent of NPF was about 100 km or larger during the EUCAARI campaign in 2008
(Kerminen et al., 2010), while Wehner et al. (2007) estimated a corresponding scale of up to 400 km
with clear horizontal variability in NPF characteristics during the SATURN campaign in 2002. The
number concentrations and size distributions of naturally charged particles (air ions) were under
investigation during EUCAARI-LONGREX campaign in May 2008 (Mirme et al., 2010). They
reported that NPF takes place throughout the whole BL, and that the particles have formed more
likely via neutral than ion-induced pathways inside the PBL.
In addition to NPF near to the surface inside the PBL and NPF in the free troposphere (FT) (Bianchi
et al., 2016), NPF has also been observed near clouds (Wehner et al., 2015). Siebert et al. (2004) and
Platis et al. (2016) observed NPF to initiate on top of the boundary layer in a capping inversion
followed by subsequent mixing of the freshly formed particles throughout the well-mixed boundary
layer. Similar observations were reported by Chen et al. (2017). Wehner et al. (2010) studied NPF in
the residual layer and observed that turbulent mixing is likely to lead to a local super saturation of



possible precursor gases, which is essential for NPF. The particles were formed in parts of the residual
layer and subsequently entrained into the BL where they were detected at the surface.
NPF events are frequently occurring over the boreal forest region in Southern Finland (Kulmala et
al., 2001; Dal Maso et al., 2005; Kulmala et al., 2013). In addition to ground-based measurements at
the SMEAR II station (61°51'N, 24°17'E, 181 m above sea level, Hari and Kulmala, 2005), which
have been conducted continuously since 1996, also airborne measurements of aerosol particles have
been carried out near the station since the year 2003 during several campaigns using a small aircraft
(O'Dowd et al., 2009; Schobesberger et al., 2013; Väänänen et al., 2016) and a hot-air balloon
(Laakso et al., 2007). Laakso et al. (2007) observed NPF to occur in the mixed BL, but also in the FT
with no connection to the BL nucleation. O'Dowd et al. (2009) observed NPF throughout the BL over
the SMEAR II, with the nucleation mode number concentration peaking first above the forest canopy.
Schobesberger et al. (2013) observed NPF inside the PBL. High concentrations of nucleation mode
particles were also found in the upper parts of the PBL, which indicates that nucleation does not
necessarily occur only close to the surface. Väänänen et al. (2016) studied the vertical and horizontal
extent of NPF in the lower troposphere near to the SMEAR II station. They observed that the air
masses within 30 km from SMEAR II differed only slightly from the ground-based observations at
the station, although the variability was larger for nucleation mode particles than for larger particles.
Furthermore, Väänänen et al. (2016) detected NPF to take place both inside the BL and, occasionally,
in the FT.
One of the sinks of newly formed aerosol particles in the PBL is dry deposition, which is important
especially for the smallest particles (Rannik et al., 2000; Lauros et al., 2011). Recently, Zha et al.
(2017) studied the vertical profile of highly oxygenated organic compounds (HOMs), which are
known precursors for aerosol formation (Ehn et al., 2014). They found that while the concentrations
were similar below and above canopy (35 m) during well-mixed conditions, the concentrations were
often clearly lower near the ground level during night-time, when temperature inversion occurred,
probably due to changes in their sources and sinks (e.g. surface deposition) during stable conditions.
In this study, we investigate the vertical variation of 1.5–3 nm and 3–10 nm particles from the ground
level up to 3 kilometres during different kind of days in relation to the occurrence of NPF at the
ground level, as well as the vertical mixing of a particle population within the evolving BL. The
dataset was collected during three measurement flight campaigns, in spring 2015, August 2015 and
in spring 2017, within a 30-km distance from the SMEAR II station. The results are compared to the



data measured on the ground level at the station. Traditional NPF event classification is used to
classify studied days as NPF events, non-events and undefined days (Dal Maso et al., 2005).
The questions we would like to answer are: Which kind of characteristics do we have in the vertical
profile of small particles?; How do these profiles differ between the NPF event, non-event and
undefined days?; Where do new particles form and how does the strength of turbulent mixing affect
particle concentrations?; What is the median concentration of small particles inside the BL during the
NPF event, non-event and undefined days, and how well do the results agree with the values measured
on the ground level?

## 100  2  Materials and methods

### 101  2.1  Measurements on board Cessna

As a platform for aerosol instruments, we used a light one-engine Cessna FR172F aircraft. The
measurement instruments were installed on an aluminium rack at the middle part inside the plane's
cabin (Fig. 1). A steel inlet line (with 32 mm inner diameter) was mounted onto the top of the rack
and lifted in and out from the window in the left side of the plane. The sample was collected from a
50-cm distance from the fuselage of the plane. The main flow in the steel tube was kept constant at
47 l min$^{-1}$ during the measurement flight and was produced by suction in the venturi and forward
motion of the airplane. Each instrument took their actual inlet flow from the central line of the main
flow, minimizing the diffusional losses of the smallest particles. The measurements were performed
with an airspeed of 125 km/h. More details about partly the same instrumentation and layout can be
found in Schobesberger et al. (2013) and Väänänen et al. (2016). The data were collected within a
30-km distance from SMEAR II station and the area is covered mainly by coniferous forest.

### 113  2.1.1  Instrumentation

The main instrumentation for this study consisted of several different particle counters. An ultrafine
condensation particle counter (uCPC, model TSI-3776) is an instrument that detects the total
concentration of particles larger than about 3 nm in diameter. Particles larger than the threshold
diameter are grown into large droplets by condensing butanol vapour onto their surface, after which
they are detected optically with a laser-diode photodetector. The ultrafine CPC has an internal vacuum
pump that draws the aerosol sample with flow rate of 1.5 l min$^{-1}$ into the instrument.




Airmodus Ltd has developed a mixing-type Particle Size Magnifier (PSM). The instrument is able to
detect directly sub-3 nm atmospheric particles using diethylene glycol (DEG) as condensing fluid
(Vanhanen et al., 2011). Compared with typically-used working fluids in CPCs, water and butanol,
the advantages of using DEG as condensing fluid are its lower saturation vapour pressure and higher
surface tension, which enables to detect particles down to 1 nm. The PSM requires a separate water
or butanol counter (CPC) for detecting optically the grown particles. The PSM in this study was a
model A10, operating with a butanol CPC (model TSI-3010). During the flight measurements
presented here, the instrument was used in fixed saturator flow rate mode measuring the total particle
concentration with a 1.5 nm cut-off size.
The instrumentation included also a custom-built Scanning Mobility Particle Sizer (SMPS), which
measures the particle number size distribution in the diameter size range of 10–400 nm with a 2-min
time resolution. Before the classification of an aerosol population, the particles are transported to a
radioactive source where they reach a constant bipolar charge equilibrium. The SMPS contains a
differential mobility analyser (DMA, Hauke type), while particle number concentrations are
measured with a butanol CPC (model TSI-3010).
The concentrations of water vapour ($H_2O$) and carbon dioxide ($CO_2$) were measured with a Li-Cor
(LI-840) gas analyser located in the instrumentation rack. Basic meteorological variables, including
the ambient temperature, relative humidity (RH) and static pressure, were measured. Pressure was
measured inside the plane while the temperature and RH sensor was located in the right wing of the
plane. The location of plane was recorded by a GPS receiver.
**2.2   SMEAR II research station**
A research Station for Measuring Ecosystem-Atmospheric Relations (SMEAR) II in Hyytiälä,
Southern Finland, was established in 1995 (see Hari and Kulmala, 2005). The station is equipped
with several aerosol and gas instruments together with flux, irradiation and meteorological
measurements. The long-term measurements give reliable and comprehensive knowledge about
ambient conditions at a relatively clean coniferous forest site. The station includes ground-based
measurements, tower measurements at the 35-m height above the ground level right above the
canopy, and measurements conducted from a mast at different altitudes up to 128 m.
In this study, we mainly used particle data from the ground level as a reference data to which we
compare our flight measurement data. The number concentrations in the size range of 1.5–3 nm were
calculated from the difference between the measured total particle concentration at 1.5 nm cut-off



size (from the PSM) and total concentration at 3 nm cut-off size (from DMPS). The distance between
the PSM and DMPS is vertically a few meters and horizontally a few tens of meters, which causes
some uncertainties in 1.5–3 nm particle number concentrations, especially during poorly-mixed BL
times in the morning when the two instruments do not always measure the same air mass.
The sensible heat flux (SHF) was measured at the at 23-m height, and we used these data to get
qualitative information on the strength of vertical mixing in the measured air masses.

### 2.3   Data analysis

The particle number concentration in size range of 1.5–3 nm was calculated as the difference of the
total particle concentrations measured with the PSM and uCPC on board the Cessna. The cut-off sizes
of these instruments were 1.5 nm and 3 nm. The cut-off size of the SMPS was 10 nm. The number
concentration in the size range of 3–10 nm was calculated as the difference in the total particle number
concentrations measured with uCPC and SMPS.
Total particle number concentrations measured on board the Cessna were first converted into standard
temperature and pressure conditions (273.15K, 1 atm) and then were corrected with the maximum
detection efficiency of the instrument based on laboratory calibrations. The maximum detection
efficiency of the PSM used in airborne measurement was 0.75 and that of uCPC was 0.99. The
maximum detection efficiencies of the PSMs used at the station were 0.8. Finally, the particle number
concentrations were corrected with respect to diffusional losses in the inlet part (Fig. 1) and inside
the sampling lines on the plane. The ground and tower data were assumed to have negligible inlet line
losses because of core sampling (Kangasluoma et al., 2016). The correction factor for the inlet part
was 0.716 for 1.5–3 nm particles and 0.720 for 3–10 nm particles based on simulation results using
COMSOL Multiphysics. Penetration efficiency through the sampling lines in the size range of 1.5–3
nm was 0.70 and in the size range of 3–10 nm 0.88.
All the results presented here are reported vertically as meters above the ground level, and all the data
were collected from within a distance of 30 km from the SMEAR II station in Hyytiälä. A typical
measurement flight includes a linear ascent from 100 m (a.g.l.) up to the FT region, 2500–3500 m,
and a descent back near to the canopy top level.
In this study, we analysed altogether 53 measurement profiles during 18 days. The flights were
conducted during three measurement campaigns: May-June 2015, August 2015 and April-May 2017,
either in the morning (7:00–12:00, UTC+2) or in the afternoon (12:00–15:00) time. The days were



classified as event, non-event or undefined days based on the NPF event classification method by Dal
Maso et al. (2005).
Well-mixed boundary layers are capped by a stable layer. The boundary layer height (BLH) was
visually estimated for each vertical measurement profile based on the particle number concentrations,
$H_2O$ and $CO_2$ concentrations, potential temperature and relative humidity. When the sun is rising, the
mixing of air mass starts from near the ground, and aerosol particles originating from surface get
mixed upwards within the rising mixed layer. Inside the mixing layer, higher concentrations of $H_2O$
are sometimes seen when the turbulence mixes up the moisture from the surface. $CO2$ tends to be
higher in the morning boundary layer due to respiration and decreases in the residual layer. The
vertical profile of the potential temperature is almost constant in the surface mixed layer and rapidly
increases with an increasing altitude under stable conditions.

## 2.4  Uncertainties


As described above, all the results were converted into STP-conditions and corrected for the
instrumental maximum detection efficiency and line losses according to the laboratory
characterizations of the flight setup. However, there are several factors causing uncertainties in the
measured concentrations. The flight speed, main flow rate, air pressure, relative humidity and
temperature are changing rapidly during a flight, which can cause variations in the inlet flows and the
performance of the instruments. It is poorly known how the uCPC and PSM behave under quickly
varying operational conditions. The reduced pressure at high altitudes may change the maximum
detection efficiency and cut-off size of laminar flow CPCs (e.g. Zhang and Liu, 1991; Herrman and
Wiedensohler, 2001). The pressure effect on the PSM cut-off size has been observed to be small (<
0.1 nm until 60 kPa) compared to the uncertainty caused by a changing relative humidity and particle
composition (Kangasluoma et al., 2016). Because of the uncertainties in the instrument cut-off sizes,
the true size range of the 1.5–3 nm concentration may vary with altitude and between different flights.
Because of the uncertainties in the determined concentrations, we should focus on the relative
behaviour of median values rather than absolute concentrations.





## 3   Results and discussion

The flight days were divided into event, non-event and undefined days based on the NPF event classification by Dal Maso et al. (2005). Based on this classification on the ground level, the vertical profiles of particles in the size ranges of 1.5–3 nm and 3–10 nm were studied separately in each type of days. During event and undefined days, we also looked at differences between the morning and afternoon times. The number of flights during non-event days is low (two vertical profiles), because cloudiness makes the operation of the aircraft impossible. Non-event days are mostly cloudy in Hyytiälä (Dada et al., 2017).

For the flight days, when we have comparable particle data from the ground station, we calculated the median values of 1.5–3 nm particle concentration both inside BL on board the Cessna and on the ground level. The boundary layer height was estimated for every vertical measurement profile.

### 3.1   General features and vertical profiles

The median values of particle concentrations, sensible heat flux (SHF) and estimated BLH was calculated for the 27 cases when comparable data were available at the SMEAR II station (Table 1). The values inside BL indicates here the observations on board Cessna, which means that the minimum limit for altitude was around 100 m from ground level. The values on the ground level were measured inside the forest canopy.

On average, we found that the concentration of 1.5–3 nm particles were higher inside the BL (1400 $cm^{-3}$) than on the ground station level (1100 $cm^{-3}$) (referred to from here as 'ground'). The values were the highest on NPF event days (1500 $cm^{-3}$ inside BL and 1300 $cm^{-3}$ on the ground) and undefined days (1450 $cm^{-3}$ inside BL and 1130 $cm^{-3}$ on the ground) and clearly the lowest on non-event day (890 $cm^{-3}$ inside BL and 740 $cm^{-3}$ on the ground) both inside the BL and on the ground level. It should be noted that both of two non-event profiles were measured during the same afternoon in the spring of 2015.

The observation of having somewhat lower concentrations of small particles at ground level is probably due to higher sinks of particles and their precursors inside the canopy compared with above-canopy air (Zha et al., 2017).

The median BLH of all the profiles was 1400 m, being lower in the morning (1100 m) and higher during the afternoon flights (2000 m). Indicative of stronger vertical mixing, the median value of the



sensible heat flux (SHF) was the highest on the NPF event days, especially during the afternoon (286
W m$^{-2}$).
Figure 2 shows the median vertical profiles of the total particle number concentration in the size
ranges of 1.5–3 nm and 3–10 nm separately for the NPF event days, undefined days and one non-
event day. The profiles typically contain data from 100 m up to 2700 m above the ground level. It is
noticeable that non-event profile consists only two vertical profiles and both of them were measured
in the same afternoon. We found that airborne 1.5–3 nm particle concentrations were similar between
the event and undefined days, whereas substantially lower concentrations were observed on non-
event day. We also observed that during the event days there were clearly more 3–10 nm particles
inside BL than during undefined days (Fig. 2a and 2b). The reason for this could be that during the
undefined days the formation of sub-3 nm particles took place, yet the conditions were not suitable
for the particle growth to larger sizes (see Buenrostro Mazon et al., 2009; Kulmala et al., 2013). Our
findings are consistent with earlier observations of high sub-3 nm particle concentrations in Hyytiälä
on both event and undefined days compared with non-event days (Lehtipalo et al., 2009; Dada et al.,

2017).

During the NPF event days, median, 25$^{th}$ and 75$^{th}$ percentiles show that the concentration of sub-3
nm particles was relatively the highest right above the canopy top. This indicates that the sources of
particles and their precursor vapors are near the ground level. During the undefined days, the origin
of sub-3 nm particles was not necessarily at the ground level, as their concentration decreased right
before the ground level (from 100 m to 200 m). In addition, reviewing the median values in Table 1,
the concentration of 1.5–3 nm particles was observed to be higher inside the BL during morning times
of undefined days (2800 cm$^{-3}$) than during afternoon times (1150 cm$^{-3}$), oppositely to event days
(1070 cm$^{-3}$ and 3020 cm$^{-3}$, respectively), which supports this hypothesis (Table 1). The concentrations
of both sub-3 nm and 3–10 nm particles were very low during the non-event days and we did not
observe any clear layers for these particles. However, it should be noted that our study included only
two such profiles, since the flight measurements were not possible to conduct during non-event days
due to meteorological conditions, especially cloudiness.
The measurement flights were conducted either in the morning (7:00–12:00, UTC+2) or in the
afternoon (12:00–15:00). We studied the median vertical particle concentrations separately for those
two times in order to estimate the effect of mixing strength on the vertical profile of particles on NPF
event and undefined days. As expected based on observed SHF fluxes, we found that the
concentrations of 1.5–3 nm particles inside the BL were, on average, most homogenous vertically
during the afternoons of the NPF event days (Fig. 3).





On NPF event days, we can see an interesting layer of 3–10 nm particles in the morning above the
BL at 2400 m. From this layer, the particles can mix down into the evolving BL. Similar behavior is
seen also on undefined days, when the increase in concentration of 1.5–3 nm particles is observed in
layer right below 2500 m in the morning and the particles are grown in size and mix downward until
afternoon.
**3.2   Diurnal variation of particle concentration at different altitudes in the lower**
**atmosphere**
We studied the median diurnal variation of total particle concentration (all particles > 1.5 nm) and
separately particle concentration in size range of 1.5–3 nm at different altitudes from around 100 m
to 2700 m above the ground level around the SMEAR II station area. The study included 17 vertical
measurement profiles during event days and 34 during undefined days. From Fig. 4a it can be seen
that the total particle number concentration over all measurement profiles was the highest near the
ground in the morning. The aerosol population mixed with cleaner air within the evolving BL after
the morning, which led to a decreasing particle number concentration, whereas the concentration
increased again towards the afternoon, presumably as a result of NPF. The highest particle number
concentrations were observed at 11:30–14:30 inside the BL, which coincides with the peak time of
NPF in Hyytiälä (Dada et al., 2018, in Prep.).
The sub-3 nm particle number concentrations (Fig. 4b) were the highest in the morning near the
ground level, with a second maximum around the noon. Later in the afternoon, sub-3 nm particle
concentration was clearly lower, probably because they apparently grew efficiently to larger sizes
and contributed significantly to the total particle concentration (Yli-Juuti et al., 2011). Both total
particles and sub-3 nm particles had the highest concentrations near the ground level throughout the
day, even though especially the total particle population seems to have been spread within the whole
mixed layer.
Figure 4c show the data availability for this analysis. It is noticeable that the number of data in each
100 m-half-an-hour cell varies considerably. In addition, one intense NPF event day with strong
particle formation in the early morning dominated the distribution due to the low number of flights at
around 7:00–8:00. Most of the data were collected either during the morning (8:30–11:30) or
afternoon (13:30–15:00). As we know, also the BLH, mixing of air and meteorological conditions
can differ significantly even within one day, and especially so between the NPF event and undefined
days.




### 3.3   Case study – NPF in evolving BL


The 13th of August 2015 was an intense NPF event day in Hyytiälä (Fig. 5a). During that day we
conducted two measurement flights around the SMEAR II station and observed the particle
concentration in size range of 1.5–3 nm to follow the development of BL and turbulent mixing (Fig.
6a, 6c, 7a, 7c). During the first measurement flight at 7:30–9:00, we observed a clear layer of 3–10
nm particles near the FT region above 2300 m. These particles were mixed down before the afternoon
flight, as this population was not anymore observed during that flight. The negative (downwards)
particle flux at SMEAR II after 12:00 supports this hypothesis (Fig. 5b).
The estimated BLH was ~700 meters during the first flight in the morning and had risen up to 1500–
1700 meters until afternoon flight.  Below the FT, the vertical variation of the 1.5–3 nm particle
concentration was larger compared to the stable conditions in FT. The concentration of 1.5–3 nm
particles inside the BL increased during the morning flight (Fig. 6a and 6c) and decreased during
afternoon flight (Fig. 7a and 7c), whereas 3–10 nm particles seemed to behave in an opposite manner.
The sub-3 nm particle concentrations were clearly higher inside the BL than in the FT, and the
concentration increased towards the ground. This is consistent with organic vapors, emitted from the
ground vegetation, participating in NPF and growth (Kulmala et al., 2013; Ehn et al., 2014).

## 4   Conclusions



Small 1.5–3 nm particles were observed inside the convective BL on-board a Cessna aircraft. On
average, the highest concentrations of sub-3 nm particles were found during NPF event mornings
above the forest canopy top. This points towards the forest being an important source of the precursor
vapors for newly formed particles. Due to the convective mixing inside BL, small particles near the
ground started to mix up while sub-10 nm particles mixed down from the FT region. Strong vertical
mixing was more typical for the NPF event days than for the undefined and non-event days, especially
during the afternoon. The concentration of sub-3 nm particles was clearly higher inside the BL on
both NPF event days and undefined days compared with one non-event day, but their vertical
variation was somewhat different, reflecting the different mixing conditions. The event days also
showed a clear increase of 3–10 nm particles in the afternoon, which was missing on undefined days
when the NPF process had been interrupted.



We found that airborne and on-ground median concentrations of sub-3 nm particles were mostly in
good agreement. Some differences still existed, which can be explained by poor vertical mixing of
air, changes in air mass origins and regional variations. The concentrations of sub-3 nm particles on
the ground were, on average, somewhat lower than airborne observations, which indicates a higher
sink for these particles inside the forest canopy.

**Acknowledgements**
This work was supported by the European Research Council via ERC-Advanced Grant ATM-GTP
(742206), the European Commission via projects H2020-INFRAIA-2014-2015 project ACTRIS-2
(Aerosols, Clouds, and Trace gases Research InfraStructure), H2020 research and innovation 345
programme under grant agreement No 689443 (ERAPLANET) via project iCUPE (Integrative and
Comprehensive Understanding on Polar Environments), FP7 project BACCHUS (Impact of Biogenic
versus Anthropogenic emissions on Clouds and Climate: towards a Holistic UnderStanding, FP7-
603445), Academy of Finland via Centre of Excellence in Atmospheric Sciences (272041) and
NanoBiomass (307537).

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





**Tables**

Table 1. Numerical statistics about boundary layer height (BLH) and sensible heat flux (SHF) indicating the mixing of air mass, and concentrations of 1.5–3 nm particles during measurement flights in 2015 and 2017. The morning flights have been conducted between 7:00–12:00 o'clock and afternoon flights at 12:00–15:00 o'clock. The low number of flights during non-event days is caused by the cloudiness which makes the operation of the aircraft impossible.

|  | Number of flight profiles | Median conc. (1.5–3 nm) inside BL [cm$^{-3}$] | Median conc. (1.5–3 nm) on ground level [cm$^{-3}$] | Median BLH [m] | Median SHF [W m$^{-2}$] |
|---|---|---|---|---|---|
| All days | 27 | 1404 | 1104 | 1400 | 192.3 |
| morning | 13 | 1995 | 888 | 1100 | 174.6 |
| afternoon | 14 | 1232 | 1251 | 2000 | 220.5 |
| Events | 11 | 1509 | 1300 | 1250 | 200 |
| morning | 6 | 1066 | 950 | 800 | 154.5 |
| afternoon | 5 | 3019 | 1435 | 1550 | 285.8 |
| Undefined | 14 | 1450 | 1129 | 1450 | 180.7 |
| morning | 7 | 2793 | 838 | 1200 | 182.6 |
| afternoon | 7 | 1149 | 1169 | 2000 | 178.7 |
| Non-events | 2 | 887 | 744 | 2000 | 162.3 |
| morning | - | - | - | - | - |
| afternoon | 2 | 887 | 744 | 2000 | 162.3 |



**Figures**

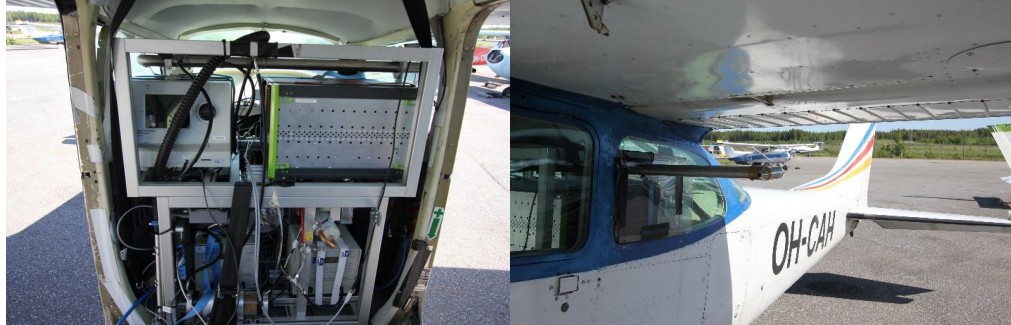



Figure 1. Instrumentation rack was installed inside the cabin (on the left) and the sample air for the
instrumentation was taken from a steel tube at 50 cm distance from the fuselage of the plane (on the
right).

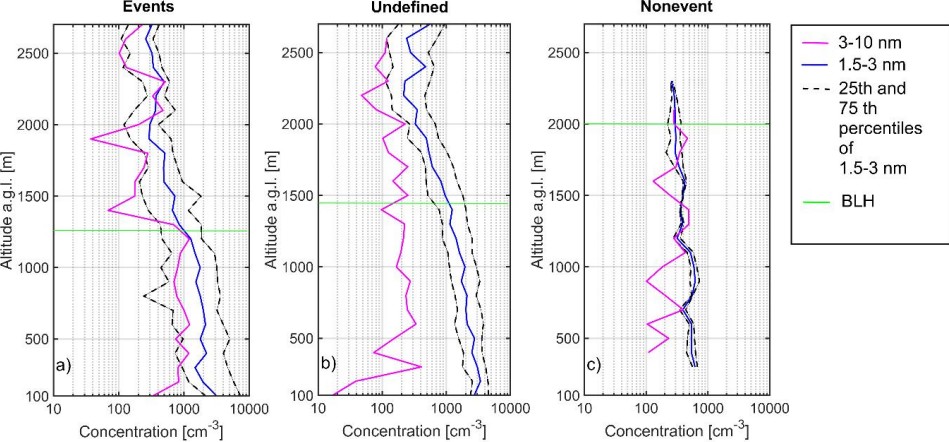



Figure 2. All day median particle concentrations in two size ranges, 3–10 nm (pink) and 1.5–3 nm
(blue) and 25- and 75-percentiles (dashed lines) of the 1.5–3 nm particle concentration, as a
function of altitude over 17 event day (a), 34 undefined day (b) and 2 non-event day afternoon
profiles (c). The concentrations were calculated from the differences between three instruments
(PSM, uCPC and SMPS) at different cut-off sizes: 1.5 nm, 3 nm and 10 nm, respectively. The data
were collected from near (< 30 km) to SMEAR II station during spring and August flight
measurement campaigns in 2015 and spring campaign 2017. Median boundary layer heights are
marked by green lines.



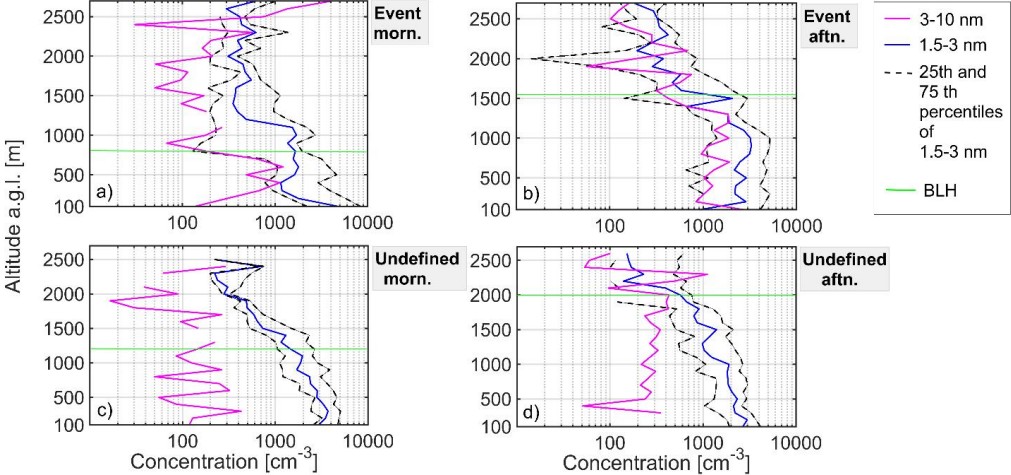



Figure 3. Median concentrations in two size ranges (1.5–3 nm and 3–10 nm) and 25- and 75-
percentiles of 1.5–3 nm particle concentration over measurement profiles during event and
undefined days separately for morning (a, c) (7:00–12:00 o'clock) and afternoon (b, d) (12:00–
15:00 o'clock) times. The median vertical profiles were defined over 9 event morning, 8 event
afternoon, 18 undefined morning and 16 undefined afternoon profiles. Median boundary layer
heights are marked by green lines.











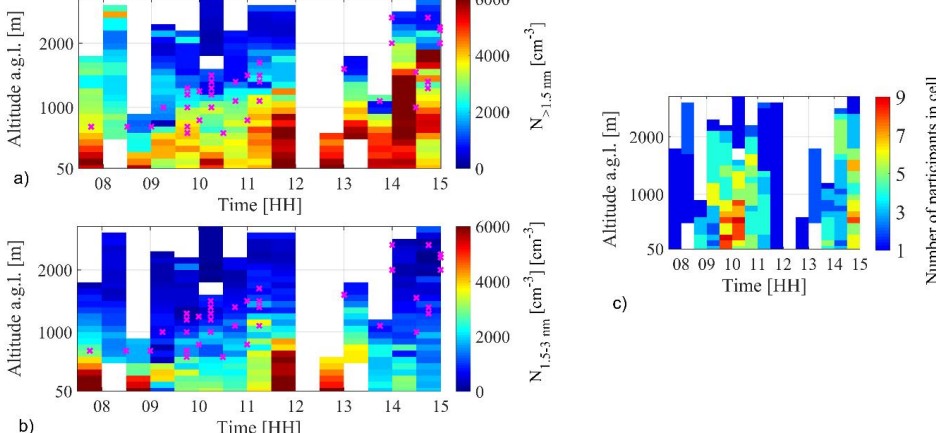



Figure 4. Panel a) shows median total particle number concentration at different altitudes calculated
over 51 measurement flight profiles (17 event day and 34 undefined day profiles) during 2015
spring and August and 2017 spring campaigns in 30 km maximum distance from SMEAR II station.
The total particle number concentration was measured with PSM with the cut-off size of 1.5 nm.
Colour scale indicates total number concentration. Panel b) shows median particle number
concentration in the size range of 1.5–3 nm at different altitudes. The value is defined as difference
of total number concentrations with different cut-off sizes; PSM (1.5 nm) and uCPC (3 nm). Panel
c) shows the number of data points in each cell of figures a-b). Estimated boundary layer heights are
marked as crosses in figures a-b) over flight profiles. Each cell includes the median value of all
measurement points inside the 100 m bin and half-an-hour.







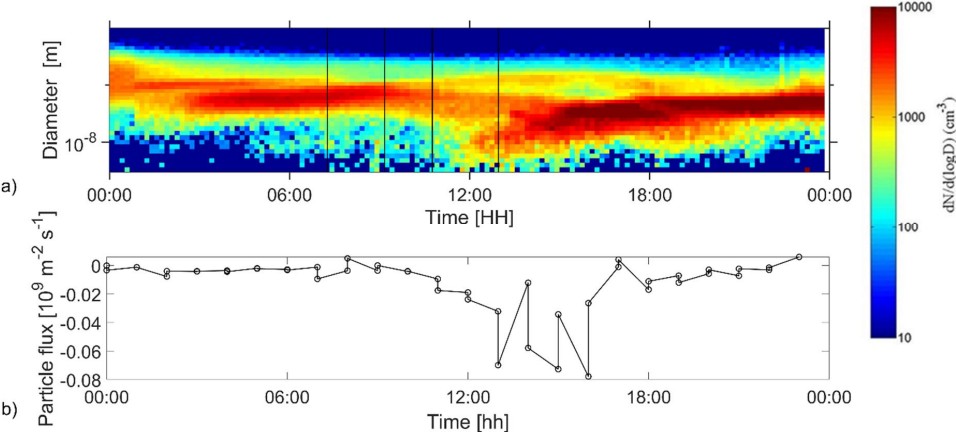


Figure 5. New particle formation event at SMEAR II station in Hyytiälä on 13$^{th}$ August 2015. Panel a) shows the number size distribution measured by Differential Mobility Particle Sizer at ground level inside the forest canopy. Start and end times of two measurement flights were marked by vertical lines in figure. Panel b shows the particle flux measured at 23 m above ground level at the station. Negative particle flux indicates particles flux downwards.

555

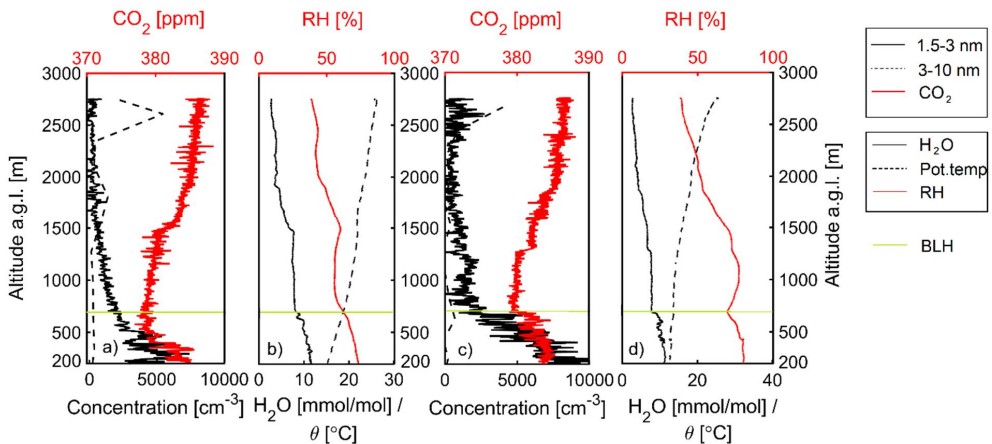

556

Figure 6. Vertical profiles during the first measurement flight at 7:30–9:00 a.m. on 13$^{th}$ August 2015 (marked in Fig 5). Panels a, b) show data from the ascent and c, d) from the descent. Figures a) and c) show the number concentration of 1.5–3 nm (black solid line) and 3–10 nm (dashed line) particles and the carbon dioxide concentration (red). Panels b) and d) show water vapor concentration (black), relative humidity (red) and potential temperature (dashes line) profiles. The green line is the estimated boundary layer height.

563





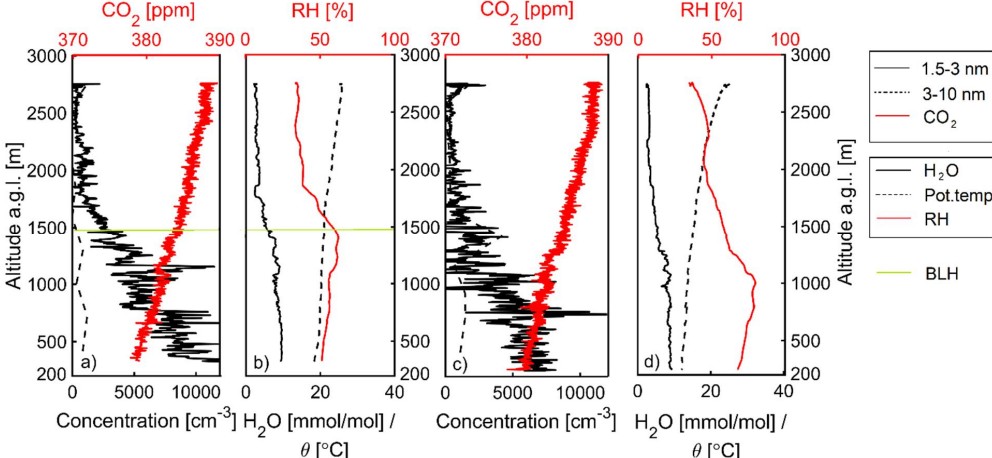

564

Figure 7. Measurement profiles like in the previous figure, but during the second measurement flight on 13[th] August 2015 at 11:00 a.m. – 12:45 p.m.