# Peer review of "Vertical profiles of sub-3 nm particles over the boreal forest"

_Atmospheric Chemistry and Physics, 2018_

## Referee Comment (RC1) · Anonymous Referee #1 · 7 Oct 2018

This paper presents airborne observations of sub-3 nm particles in the lower troposphere over the boreal forest of Finland. The results show that the number concentration of sub-3 nm particles was highest near the forest canopy top indicating the key role of the precursor vapors emitted by the forest during new particle formation (NPF). Case study shows the number concentrations of sub-3 nm particles are influenced by the evolving of boundary layer during the NPF. Overall, this study presents interesting results regarding the vertical profile of new particle formation. The manuscript is concisely organized and well written. Therefore, I suggest that this manuscript can be considered for publication after following comments are well addressed:

Specific comments:

1. As only three fights were analyzed in this study, case studies should be done for

all the three flights. In the manuscript, only 13th of August 2015 was chosen for case study. What the aerosol size distributions on ground and what the values on the aircraft were related to the values on the ground during undefine day and non-event day are also interesting to be known. Is it possible to use some other methods, such as modelling method (i.e. simulations by regional model) in case studies?

2. Can other vertical observations, such as lidar data, satellite data etc. support your study?

3. Some implications need to be added in the conclusion or even in the abstract. For example, how does this study improve the recent knowledge of NPF study? What are the highlights of this study? Why do we need to do the vertical observations? What else is needed in future?

Minor comments:

P2, L41-42: This sentence is not clear and need to be rewritten. BLH is not process.

P2, L58: What kind of observations reported by Chen et al. (2017) need to be described. If it is same with observations by Siebert et al. and Platis et al., merge these two sentences.

P3, L79-85: This paragraph is a little bit abrupt here and need to be moved to somewhere above. Maybe put it after the third paragraph.

P5, L136-137: The instrument used to measure the meteorological variables need to be described here.

Figure 1: The A11 manual said the CPC should be placed on a higher level than the PSM outlet. From the left panel of Fig. 1, it looks like the CPC is below the PSM. I wonder if it will influence the operation or observation accuracy of PSM. Moreover, some text or label can be added in the Figure. For example, add the names of each instrument in the left panel of Fig. 1 and mark the direction of the inlet in the right panel.

P6, L171-172: A citation or explanation is needed for 'COMSOL Multiphysics'. Is it a software or what? Explain the acronym once.

P7, L194: Explain the acronym once.

P8, L226-229: The expression of 'the values inside the BL' is not suitable as 'the Ground level' is also 'inside BL'.

Table 1: Give the median of height for the observations by airborne.

P8, L236-238: Merge this paragraph with previous paragraph. The guessed explanation of observed phenomenon should be right after the expression of phenomenon (i.e. after P8, L229-230.).

P9, L260: 'above the ground level'?

P10, L274-278: Why was a layer of 3-10 nm particles observed? Is it related to the origin of air mass? Section 3.3 what are the main conclusions or findings through the case study?

―――――――――――――――

---

## Referee Comment (RC2) · Anonymous Referee #3 · 6 Jan 2019

*Summary:*

This work demonstrates the Vertical profiles of sub-3 nm particles over the boreal forest. The data is valuable and the manuscript fits well to the scope of ACP. I recommend it to be published after the following comments have been adequately addressed.

*Comments:*

1. Line 16-17: The number of flight/vertical profiles is confused. There are only 13 morning flight profiles shown in Table 1, even though both the ascent and the descent flights are counted, how could be 27 morning vertical profiles in total? Please check your data.

2. Line 73-78: It seems that the vertical profiles of NPF/aerosol number size distribution around SMEAR II station have been reported (Väänänen et al., 2016), although the paper is still under discussion. I would suggest the authors to compare with the previous results.

3. Line 170-173: Does this mean the constant factor is used to correct diffusional loss for a certain size range (1.5-3 nm or 3-10 nm)? The diffusional loss for small particles should be size dependence. This method will introduce the additional uncertainty. Please clarify.

4. Line 183-185: Please explain more about the method to estimate the BLH. Is there any other vertical measurement, such as lidar, can be used?

5. Line 202: how about the pressure effect of UCPC?

6. Line 236-238: Here I would suggest the vertical profiles of condensation sink should be calculated with SMPS data, and then compared with that of ground measurements. In previous work (Zha et al., 2017), the vertical measurements were only conducted at ~36 m and ~1.5 m above ground. This height is too low to support your conclusion.

7. Line 271-273: how could explain the vertical profiles of 1.5-3 nm particles under BLH for undefined day in Fig.3? Why it is different from the NPF day?

8. Line 316-318: Please provide the precise value to support your statement.

---

## Author Comment (AC1) · 17 Feb 2019

Our response to referee comments

Referee #1

Specific comments:

1. As only three fights were analyzed in this study, case studies should be done for all the three flights. In the manuscript, only 13th of August 2015 was chosen for case study. What the aerosol size distributions on ground and what the values on the aircraft were related to the values on the ground during undefine day and non-event day are also interesting to be known. Is it possible to use some other methods, such as modelling method (i.e. simulations by regional model) in case studies?

[Figure]

Answer: In this article, we used the data from three measurement campaigns, consisting of altogether 53 individual flights, which all were included in the analysis, and one of them was chosen for a case study. For the flights studied in Figs. 2 and 3, we have the corresponding data from the ground level only for ~50% of the flights, but the ground level – boundary layer –comparison is done in Table 1, also for undefined and non-event days. For the case study day (13th of August 2015), we do not have unfortunately the data for the size range of 1.5–3 nm on the ground level, but the DMPS size distribution is shown below as Fig 1 (Fig. 5 in the article) and the flight times (two measurement flights were done that day) were marked as black vertical lines on the figure.

There are some previous modelling studies about NPF in the boundary layer above Hyytiälä. Boy et al. (2006) have modelled NPF in the lower atmosphere and vertical profiles of small particles in mixing BL at SMEAR II station in Hyytiälä. They predicted a maximum of newly formed clusters and particle concentrations near the ground level. Lauros et al. (2011) investigated particle fluxes and deposition to evaluate different particle formation mechanisms with model simulations at SMEAR II and suggested that organic compounds emitted by the forest have a significant role in aerosol formation. These studies thus support the conclusions of this study and we feel that adding a modelling component to the case study would not bring additional new information. We added to the text: "The vertical profiles of small particles in mixing BL at SMEAR II were modelled by Boy et al. (2006). The results gave the maximum of newly formed clusters and particle concentrations near the ground level."

2. Can other vertical observations, such as lidar data, satellite data etc. support your study? '

Answer: Doppler lidar was operating at SMEAR-II at time of the flight campaigns, but unfortunately in the springtime clear-sky days, when most of the flights took place, sensitivity of the instrument was usually not sufficient to provide retrievals up to the top of the boundary layer. Thus, in this study the in-situ measurements onboard the

Cessna aircraft were considered to be more reliable for estimating the BLH but a visual comparison with Doppler lidar was done when possible.

3. Some implications need to be added in the conclusion or even in the abstract. For example, how does this study improve the recent knowledge of NPF study? What are the highlights of this study? Why do we need to do the vertical observations? What else is needed in future?

Answer: We agree with the referee. We added to the conclusions: "This study increases our understanding of the first steps of atmospheric NPF inside the whole BL and the connections between atmospheric mixing and NPF. Next step would be to investigate different formation pathways in more detail. To achieve this, it would be important to find out also the chemical composition of particles above the ground level so that we could assess more specifically the possible sources of the precursor gases. In addition, the contribution of mesoscale convection induced movement, like roll vortices to NPF is currently under investigation." To the abstract we added a concluding sentence: "The results shed light on the connection between boundary layer dynamics and NPF."

Minor comments:

P2, L41-42: This sentence is not clear and need to be rewritten. BLH is not process.

Answer: We will replace "Several meteorological, physical and chemical processes influence the spatial and temporal conditions inside the BL, such as the boundary layer height (BLH) and mixing strength." → "Several meteorological, physical and chemical processes influence the spatial and temporal conditions inside the BL and thus the mixing strength and evolution of boundary layer."

P2, L58: What kind of observations reported by Chen et al. (2017) need to be described. If it is same with observations by Siebert et al. and Platis et al., merge these two sentences.

Answer: Yes, we can merge these sentences. "Siebert et al. (2004), Platis et al. (2016) and Chen et al. (2017) observed NPF to initiate on top of the boundary layer..."

P3, L79-85: This paragraph is a little bit abrupt here and need to be moved to some-where above. Maybe put it after the third paragraph.

Answer: We agree that it could be better there.

P5, L136-137: The instrument used to measure the meteorological variables need to be described here.

Answer: "Basic meteorological variables, including the ambient temperature (with PT-100 temperature sensor), relative humidity (RH) (with Rotronic HygroClip-S sensor) and static pressure (with Vaisala PTB100B), were measured."

Figure 1: The A11 manual said the CPC should be placed on a higher level than the PSM outlet. From the left panel of Fig. 1, it looks like the CPC is below the PSM. I wonder if it will influence the operation or observation accuracy of PSM. Moreover, some text or label can be added in the Figure. For example, add the names of each instrument in the left panel of Fig. 1 and mark the direction of the inlet in the right panel.

Answer: The figure is from the first flight measurement campaign in 2015. In the mea-surement campaign in 2017 we placed the CPC to the top of the PSM according to the manual instructions. The placing of the instruments in this configuration is impor-tant during long-term operation for preventing possible excess droplets of DEG from the PSM entering inside the CPC. However, as the flight times were rather short, the possibility of DEG contamination is small and we did not detect any problems in the performance of the CPC.

We replaced the figure 1 with the other one (here Fig2).

P6, L171-172: A citation or explanation is needed for 'COMSOL Multiphysics'. Is it a software or what? Explain the acronym once.

[Figure]

Answer: COMSOL Multiphysics is a name of software. This is added to the text.

P7, L194: Explain the acronym once.

Answer: Standard temperature and pressure (STP, 100 kPa, 273.15K)

P8, L226-229: The expression of 'the values inside the BL' is not suitable as 'the Ground level' is also 'inside BL'.

Answer: This is true. We replaced "the values inside the BL" with "the values onboard aircraft (inside the BL).

Table 1: Give the median of height for the observations by airborne.

Answer: We added this to the table.

P8, L236-238: Merge this paragraph with previous paragraph. The guessed explanation of observed phenomenon should be right after the expression of phenomenon (i.e. after P8, L229-230.).

Answer: We agree.

P9, L260: 'above the ground level'?

Answer: Yes, above the ground level.

P10, L274-278: Why was a layer of 3-10 nm particles observed? Is it related to the origin of air mass? Section 3.3 what are the main conclusions or fidings through the case study?

Answer: The exact origin of these particles is currently a subject to another study, but we speculate that they could be related to the residual layer from the previous day. The case study supports the hypothesis about the intensive particle formation in the mixing boundary layer in early morning. The negative particle flux indicates newly formed particles mixing down into the canopy that could explain the observations of onset of NPF later in the ground level.

Referee #3:
Summary: This work demonstrates the Vertical profiles of sub-3 nm particles over the boreal forest. The data is valuable and the manuscript fits well to the scope of ACP. I recommend it to be published after the following comments have been adequately addressed.

Comments:

1. Line 16-17: The number of flight/vertical profiles is confused. There are only 13 morning flight profiles shown in Table 1, even though both the ascent and the descent flights are counted, how could be 27 morning vertical profiles in total? Please check your data.

Answer: We measured 27 morning and 26 afternoon flight profiles in total, however we have the corresponding data available from the ground level only for ca. 50% of the flight times. Therefore, in Table 1, we have selected only those flight profiles for which we have the values from the ground level as well, so that the values are comparable. For Fig. 2, the whole flight data set has been considered. This is now clarified in the Table 1 label.

2. Line 73-78: It seems that the vertical profiles of NPF/aerosol number size distribution around SMEAR II station have been reported (Väänänen et al., 2016), although the paper is still under discussion. I would suggest the authors to compare with the previous results.

Answer: This is true. We referred to Väänänen et al., 2016 only in case of description of instrumentation.

3. Line 170-173: Does this mean the constant factor is used to correct diffusional loss for a certain size range (1.5-3 nm or 3-10 nm)? The diffusional loss for small particles should be size dependence. This method will introduce the additional uncertainty. Please clarify.

Answer: The referee is correct that the diffusion losses are of course in reality size dependent. However, as we only could determine the concentration in the size bins 1.5-3 nm and 3-10 nm, rather than a more detailed size distribution, we had to use one value to correct the data, which represents the average loss of the particles inside this size bin. As stated in the text, there is also uncertainty in the exact size limits of the bin, due to possible variations of the instrument cut-off size. Therefore we state: Because of the uncertainties in the determined concentrations, we should focus on the relative behaviour of median values rather than absolute concentrations. This covers both the uncertainty due to diffusion loss correction and exact size bin limits. We added a clarifying sentence about the size dependency of the losses to chapter 2.4.

4. Line 183-185: Please explain more about the method to estimate the BLH. Is there any other vertical measurement, such as lidar, can be used?

Answer: Doppler lidar was operating at SMEAR-II at time of the flight campaigns, but unfortunately in the springtime clear-sky days, when most of the flights took place, sensitivity of the instrument was usually not sufficient to provide retrievals up to the top of the boundary layer. Thus, in this study the in-situ measurements onboard the Cessna aircraft were considered to be more reliable for estimating the BLH but a visual comparison with Doppler lidar was done when possible.

5. Line 202: how about the pressure effect of UCPC?

Answer: We measured in altitudes where pressure goes down to $\sim$70 kPa, which gives uncertainty of +-5 % for the aerosol flow rate of the 3776, and thus directly to the concentrations, as shown by Takegawa et al. 2017, Fig 3.

In addition, we compared also the concentrations measured by PSM, uCPC and SMPS in the FT where the occurrence of small particles (below 10 nm, that has been under

discussion in this paper) is very low. The concentrations did match well, so we assume that the pressure effect to the measured concentrations for any of the instruments is not significant for our study.

6. Line 236-238: Here I would suggest the vertical profiles of condensation sink should be calculated with SMPS data, and then compared with that of ground measurements. In previous work (Zha et al., 2017), the vertical measurements were only conducted at ∼36 m and ∼1.5 m above ground. This height is too low to support your conclusion.

Answer: We were referring to the possible sink due to dry deposition into the forest canopy, rather than the condensation sink. The effect of canopy has been shown to be significant e.g. for small ions (Tammet et al. 2006) and possibly particle precursor vapors (Zha et al).

7. Line 271-273: how could explain the vertical profiles of 1.5-3 nm particles under BLH for undefined day in Fig.3? Why it is different from the NPF day?

Answer: There are several reasons for a day to be classified as undefined (Buenrostro Mazon et al. 2009), e.g. change of air mass, changes in cloudiness or large sink, which affect the availability and temporal evolution of the particle precursor vapors. Therefore it could be that particles start forming, but they do not continue growing, or we see only a part of the formation and growth process. The results of this study indicate that differences in the mixing conditions could also be one factor differentiating between event and undefined days.

8. Line 316-318: Please provide the precise value to support your statement.

Answer: The median of concentration of 1.5–3 nm particles inside the BL (onboard aircraft) decreased from the morning flight (7300 cm-3 during the ascent and 6300 cm-3 during the descent, Fig. 6a and 6c) to the afternoon flight (∼2500 cm-3, Fig. 7a and 7c), whereas 3–10 nm particles seemed to behave in an opposite manner (350 cm-3, 200 cm-3, 850 cm-3 and 1450 cm-3)."

Please also note the supplement to this comment:
https://www.atmos-chem-phys-discuss.net/acp-2018-605/acp-2018-605-AC1-supplement.pdf
* * *
[Figure]

**Fig. 1.**

[Figure]

**Fig. 2.**